# CayleyNets: Spectral Graph CNNs with Complex Rational Filters

## Abstract

The rise of graph-structured data such as social networks, regulatory networks, citation graphs, and functional brain networks, in combination with resounding success of deep learning in various applications, has brought the interest in generalizing deep learning models to non-Euclidean domains. In this paper, we introduce a new spectral domain convolutional architecture for deep learning on graphs. The core ingredient of our model is a new class of parametric rational complex functions (Cayley polynomials) allowing to efficiently compute spectral filters on graphs that specialize on frequency bands of interest. Our model generates rich spectral filters that are localized in space, scales linearly with the size of the input data for sparsely-connected graphs, and can handle different constructions of Laplacian operators. Extensive experimental results show the superior performance of our approach on spectral image classification, community detection, vertex classification and matrix completion tasks.

## 1 Introduction

In many domains, one has to deal with large-scale data with underlying non-Euclidean structure. Prominent examples of such data are social networks, genetic regulatory networks, functional networks of the brain, and 3D shapes represented as discrete manifolds. The recent success of deep neural networks and, in particular, convolutional neural networks (CNNs) LeCun et al. (1998) have raised the interest in *geometric deep learning* techniques trying to extend these models to data residing on graphs and manifolds. Geometric deep learning approaches have been successfully applied to computer graphics and vision Masci et al. (2015); Boscaini et al. (2015; 2016b;a); Monti et al. (2017a), brain imaging Ktena et al. (2017), and drug design Duvenaud et al. (2015) problems, to mention a few. For a comprehensive presentation of methods and applications of deep learning on graphs and manifolds, we refer the reader to the review paper Bronstein et al. (2016).

**Related work.** The earliest neural network formulation on graphs was proposed by Gori et al. (2005) and Scarselli et al. (2009), combining random walks with recurrent neural networks (their paper has recently enjoyed renewed interest in Li et al. (2015); Sukhbaatar et al. (2016)). The first CNN-type architecture on graphs was proposed by Bruna et al. (2013). One of the key challenges of extending CNNs to graphs is the lack of vector-space structure and shift-invariance making the classical notion of convolution elusive. Bruna *et al.* formulated convolution-like operations in the spectral domain, using the graph Laplacian eigenbasis as an analogy of the Fourier transform (Shuman et al. (2013)). Henaff et al. (2015) used smooth parametric spectral filters in order to achieve localization in the spatial domain and keep the number of filter parameters independent of the input size. Defferrard et al. (2016) proposed an efficient filtering scheme using recurrent Chebyshev polynomials applied on the Laplacian operator. Kipf & Welling (2016) simplified this architecture using filters operating on 1-hop neighborhoods of the graph. Atwood & Towsley (2016) proposed a Diffusion CNN architecture based on random walks on graphs. Monti et al. (2017a) (and later, Hechtlinger et al. (2017)) proposed a spatial-domain generalization of CNNs to graphs using local patch operators represented as Gaussian mixture models, showing a significant advantage of such models in generalizing across different graphs. In Monti et al. (2017b), spectral graph CNNs were extended to multiple graphs and applied to matrix completion and recommender system problems.

**Main contribution.** In this paper, we construct graph CNNs employing an efficient spectral filtering scheme based on Cayley polynomials that enjoys similar advantages of the Chebyshev filters (Defferrard et al. (2016)) such as localization and linear complexity. The main advantage of our filters over Defferrard et al. (2016) is their ability to detect narrow frequency bands of importance during training, and to specialize on them while being well-localized on the graph. We demonstrate experimentally that this affords our method greater flexibility, making it perform better on a broad range of graph learning problems.

**Notation.** We use $a$, $\mathbf{a}$, and $\mathbf{A}$ to denote scalars, vectors, and matrices, respectively. $\bar{z}$ denotes the conjugate of a complex number, $\mathrm{Re}\{z\}$ its real part, and $i$ is the imaginary unit. $\mathrm{diag}(a_1, \ldots, a_n)$ denotes an $n \times n$ diagonal matrix with diagonal elements $a_1, \ldots, a_n$. $\mathrm{Diag}(\mathbf{A}) = \mathrm{diag}(a_{11}, \ldots, a_{nn})$ denotes an $n \times n$ diagonal matrix obtained by setting to zero the off-diagonal elements of $\mathbf{A}$. $\mathrm{Off}(\mathbf{A}) = \mathbf{A} - \mathrm{Diag}(\mathbf{A})$ denotes the matrix containing only the off-diagonal elements of $\mathbf{A}$. $\mathbf{I}$ is the identity matrix and $\mathbf{A} \circ \mathbf{B}$ denotes the Hadamard (element-wise) product of matrices $\mathbf{A}$ and $\mathbf{B}$. Proofs are given in the appendix.

## 2 SPECTRAL TECHNIQUES FOR DEEP LEARNING ON GRAPHS

**Spectral graph theory.** Let $\mathcal{G} = (\{1, \ldots, n\}, \mathcal{E}, \mathbf{W})$ be an undirected weighted graph, represented by a symmetric *adjacency matrix* $\mathbf{W} = (w_{ij})$. We define $w_{ij} = 0$ if $(i, j) \notin \mathcal{E}$ and $w_{ij} > 0$ if $(i, j) \in \mathcal{E}$. We denote by $\mathcal{N}_{k,m}$ the *k-hop neighborhood* of vertex $m$, containing vertices that are at most $k$ edges away from $m$. The *unnormalized graph Laplacian* is an $n \times n$ symmetric positive-semidefinite matrix $\mathbf{\Delta}_u = \mathbf{D} - \mathbf{W}$, where $\mathbf{D} = \mathrm{diag}(\sum_{j \neq i} w_{ij})$ is the *degree matrix*. The *normalized graph Laplacian* is defined as $\mathbf{\Delta}_n = \mathbf{D}^{-1/2} \mathbf{\Delta}_u \mathbf{D}^{-1/2} = \mathbf{I} - \mathbf{D}^{-1/2} \mathbf{W} \mathbf{D}^{-1/2}$. In the following, we use the generic notation $\mathbf{\Delta}$ to refer to some Laplacian.

Since both normalized and unnormalized Laplacian are symmetric and positive semi-definite matrices, they admit an eigendecomposition $\mathbf{\Delta} = \mathbf{\Phi} \mathbf{\Lambda} \mathbf{\Phi}^\top$, where $\mathbf{\Phi} = (\phi_1, \ldots \phi_n)$ are the orthonormal eigenvectors and $\mathbf{\Lambda} = \mathrm{diag}(\lambda_1, \ldots, \lambda_n)$ is the diagonal matrix of corresponding non-negative eigenvalues (spectrum) $0 = \lambda_1 \leq \lambda_2 \leq \ldots \leq \lambda_n$. The eigenvectors play the role of Fourier atoms in classical harmonic analysis and the eigenvalues can be interpreted as (the square of) frequencies. Given a signal $\mathbf{f} = (f_1, \ldots, f_n)^\top$ on the vertices of graph $\mathcal{G}$, its *graph Fourier transform* is given by $\hat{\mathbf{f}} = \mathbf{\Phi}^\top \mathbf{f}$. Given two signals $\mathbf{f}, \mathbf{g}$ on the graph, their *spectral convolution* can be defined as the element-wise product of the Fourier transforms, $\mathbf{f} \star \mathbf{g} = \mathbf{\Phi}\big((\mathbf{\Phi}^\top \mathbf{g}) \circ (\mathbf{\Phi}^\top \mathbf{f})\big) = \mathbf{\Phi} \, \mathrm{diag}(\hat{g}_1, \ldots, \hat{g}_n) \hat{\mathbf{f}}$, which corresponds to the property referred to as the *Convolution Theorem* in the Euclidean case.

**Spectral CNNs.** Bruna et al. (2013) used the spectral definition of convolution to generalize CNNs on graphs, with a spectral convolutional layer of the form

$$\mathbf{f}_l^{\mathrm{out}} = \xi \left( \sum_{l'=1}^p \mathbf{\Phi}_k \hat{\mathbf{G}}_{l,l'} \mathbf{\Phi}_k^\top \mathbf{f}_{l'}^{\mathrm{in}} \right). \tag{1}$$

Here the $n \times p$ and $n \times q$ matrices $\mathbf{F}^{\mathrm{in}} = (\mathbf{f}_1^{\mathrm{in}}, \ldots, \mathbf{f}_p^{\mathrm{in}})$ and $\mathbf{F}^{\mathrm{out}} = (\mathbf{f}_1^{\mathrm{out}}, \ldots, \mathbf{f}_q^{\mathrm{out}})$ represent respectively the $p$- and $q$-dimensional input and output signals on the vertices of the graph, $\mathbf{\Phi}_k = (\phi_1, \ldots, \phi_k)$ is an $n \times k$ matrix of the first eigenvectors, $\hat{\mathbf{G}}_{l,l'} = \mathrm{diag}(\hat{g}_{l,l',1}, \ldots, \hat{g}_{l,l',k})$ is a $k \times k$ diagonal matrix of spectral multipliers representing a learnable filter in the frequency domain, and $\xi$ is a nonlinearity (e.g., ReLU) applied on the vertex-wise function values. Pooling is performed by means of graph coarsening, which, given a graph with $n$ vertices, produces a graph with $n' < n$ vertices and transfers signals from the vertices of the fine graph to those of the coarse one.

This framework has several major drawbacks. First, the spectral filter coefficients are *basis dependent*, and consequently, a spectral CNN model learned on one graph cannot be transferred to another graph. Second, the computation of the forward and inverse graph Fourier transforms incur expensive $\mathcal{O}(n^2)$ multiplication by the matrices $\mathbf{\Phi}, \mathbf{\Phi}^\top$, as there is no FFT-like algorithms on general graphs. Third, there is no guarantee that the filters represented in the spectral domain are localized in the spatial domain (locality property simulates local reception fields, Coates & Ng (2011)); assuming $k = \mathcal{O}(n)$ Laplacian eigenvectors are used, a spectral convolutional layer requires $\mathcal{O}(pqk) = \mathcal{O}(n)$ parameters to train.

To address the latter issues, Henaff et al. (2015) argued that smooth spectral filter coefficients result in spatially-localized filters (an argument similar to vanishing moments). The filter coefficients are represented as $\hat{g}_i = g(\lambda_i)$, where $g(\lambda)$ is a smooth transfer function of frequency $\lambda$. Applying such filter to signal $\mathbf{f}$ can be expressed as $\mathbf{Gf} = g(\boldsymbol{\Delta})\mathbf{f} = \boldsymbol{\Phi}g(\boldsymbol{\Lambda})\boldsymbol{\Phi}^\top\mathbf{f} = \boldsymbol{\Phi}\,\mathrm{diag}(g(\lambda_1), \ldots, g(\lambda_n))\boldsymbol{\Phi}^\top\mathbf{f}$, where applying a function to a matrix is understood in the operator functional calculus sense (applying the function to the matrix eigenvalues). Henaff et al. (2015) used parametric functions of the form $g(\lambda) = \sum_{j=1}^r \alpha_j \beta_j(\lambda)$, where $\beta_1(\lambda), \ldots, \beta_r(\lambda)$ are some fixed interpolation kernels such as splines, and $\boldsymbol{\alpha} = (\alpha_1, \ldots, \alpha_r)$ are the interpolation coefficients used as the optimization variables during the network training. In matrix notation, the filter is expressed as $\mathbf{Gf} = \boldsymbol{\Phi}\mathrm{diag}(\mathbf{B}\boldsymbol{\alpha})\boldsymbol{\Phi}^\top\mathbf{f}$, where $\mathbf{B} = (b_{ij}) = (\beta_j(\lambda_i))$ is a $k \times r$ matrix. Such a construction results in filters with $r = \mathcal{O}(1)$ parameters, independent of the input size. However, the authors explicitly computed the Laplacian eigenvectors $\boldsymbol{\Phi}$, resulting in high complexity.

**ChebNet.** Defferrard et al. (2016) used polynomial filters represented in the Chebyshev basis

$$g_{\boldsymbol{\alpha}}(\tilde{\lambda}) = \sum_{j=0}^r \alpha_j T_j(\tilde{\lambda}) \tag{2}$$

applied to rescaled frequency $\tilde{\lambda} \in [-1, 1]$; here, $\boldsymbol{\alpha}$ is the $(r+1)$-dimensional vector of polynomial coefficients parametrizing the filter and optimized for during the training, and $T_j(\lambda) = 2\lambda T_{j-1}(\lambda) - T_{j-2}(\lambda)$ denotes the Chebyshev polynomial of degree $j$ defined in a recursive manner with $T_1(\lambda) = \lambda$ and $T_0(\lambda) = 1$. Chebyshev polynomials form an orthogonal basis for the space of polynomials of order $r$ on $[-1, 1]$. Applying the filter is performed by $g_{\boldsymbol{\alpha}}(\tilde{\boldsymbol{\Delta}})\mathbf{f}$, where $\tilde{\boldsymbol{\Delta}} = 2\lambda_n^{-1}\boldsymbol{\Delta} - \mathbf{I}$ is the rescaled Laplacian such that its eigenvalues $\tilde{\boldsymbol{\Lambda}} = 2\lambda_n^{-1}\boldsymbol{\Lambda} - \mathbf{I}$ are in the interval $[-1, 1]$.

Such an approach has several important advantages. First, since $g_{\boldsymbol{\alpha}}(\tilde{\boldsymbol{\Delta}}) = \sum_{j=0}^r \alpha_j T_j(\tilde{\boldsymbol{\Delta}})$ contains only matrix powers, additions, and multiplications by scalar, it can be computed avoiding the explicit expensive $\mathcal{O}(n^3)$ computation of the Laplacian eigenvectors. Furthermore, due to the recursive definition of the Chebyshev polynomials, the computation of the filter $g_{\boldsymbol{\alpha}}(\boldsymbol{\Delta})\mathbf{f}$ entails applying the Laplacian $r$ times, resulting in $\mathcal{O}(rn)$ operations assuming that the Laplacian is a sparse matrix with $\mathcal{O}(1)$ non-zero elements in each row (a valid hypothesis for most real-world graphs that are sparsely connected). Second, the number of parameters is $\mathcal{O}(1)$ as $r$ is independent of the graph size $n$. Third, since the Laplacian is a local operator affecting only 1-hop neighbors of a vertex and a polynomial of degree $r$ of the Laplacian affects only $r$-hops, the resulting filters have guaranteed spatial localization.

A key disadvantage of Chebyshev filters is the fact that using polynomials makes it hard to produce narrow-band filters, as such filters require very high order $r$, and produce unwanted non-local filters. This deficiency is especially pronounced when the Laplacian has clusters of eigenvalues concentrated around a few frequencies with large spectral gap (Figure 3, middle right). Such a behavior is characteristic of graphs with community structures, which is very common in many real-world graphs, for instance, social networks. To overcome this major drawback, we need a new class of filters, that are both localized in space, and are able to specialize in narrow bands in frequency.

## 3  CAYLEY FILTERS

A key construction of this paper is a family of complex filters that enjoy the advantages of Chebyshev filters while avoiding some of their drawbacks. A *Cayley polynomial* of order $r$ is a real-valued function with complex coefficients,

$$g_{\mathbf{c},h}(\lambda) = c_0 + 2\mathrm{Re}\left\{ \sum_{j=1}^r c_j (h\lambda - i)^j (h\lambda + i)^{-j} \right\} \tag{3}$$

where $\mathbf{c} = (c_0, \ldots, c_r)$ is a vector of one real coefficient and $r$ complex coefficients and $h > 0$ is the *spectral zoom* parameter, that will be discussed later. A *Cayley filter* $\mathbf{G}$ is a spectral filter defined on real signals $\mathbf{f}$ by

$$\mathbf{Gf} = g_{\mathbf{c},h}(\boldsymbol{\Delta})\mathbf{f} = c_0\mathbf{f} + 2\mathrm{Re}\{\sum_{j=1}^r c_j (h\boldsymbol{\Delta} - i\mathbf{I})^j (h\boldsymbol{\Delta} + i\mathbf{I})^{-j}\mathbf{f}\}, \tag{4}$$

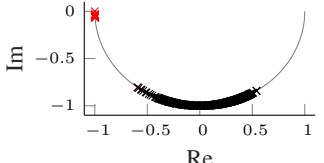 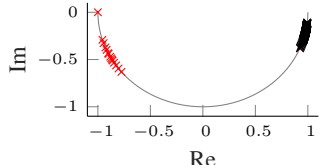 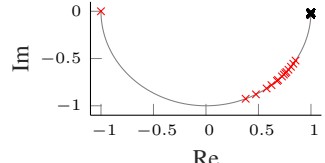

Figure 1: Eigenvalues of the unnormalized Laplacian $h\boldsymbol{\Delta}_\mathrm{u}$ of the 15-communities graph mapped on the complex unit half-circle by means of Cayley transform with spectral zoom values (left-to-right) $h = 0.1$, 1, and 10. The first 15 frequencies carrying most of the information about the communities are marked in red. Larger values of $h$ zoom (right) on the low frequency band.

where the parameters $\mathbf{c}$ and $h$ are optimized for during training. Similarly to the Chebyshev filters, Cayley filters involve basic matrix operations such as powers, additions, multiplications by scalars, and also inversions. This implies that application of the filter $\mathbf{Gf}$ can be performed without explicit expensive eigendecomposition of the Laplacian operator. In the following, we show that Cayley filters are analytically well behaved; in particular, any smooth spectral filter can be represented as a Cayley polynomial, and low-order filters are localized in the spatial domain. We also discuss numerical implementation and compare Cayley and Chebyshev filters.

**Analytic properties.** Cayley filters are best understood through the Cayley transform, from which their name derives. Denote by $e^{i\mathbb{R}} = \{e^{i\theta} : \theta \in \mathbb{R}\}$ the unit complex circle. The *Cayley transform* $\mathcal{C}(x) = \frac{x-i}{x+i}$ is a smooth bijection between $\mathbb{R}$ and $e^{i\mathbb{R}} \setminus \{1\}$. The complex matrix $\mathcal{C}(h\boldsymbol{\Delta}) = (h\boldsymbol{\Delta} - i\mathbf{I})(h\boldsymbol{\Delta} + i\mathbf{I})^{-1}$ obtained by applying the Cayley transform to the scaled Laplacian $h\boldsymbol{\Delta}$ has its spectrum in $e^{i\mathbb{R}}$ and is thus unitary. Since $z^{-1} = \overline{z}$ for $z \in e^{i\mathbb{R}}$, we can write $\overline{c_j \mathcal{C}^j(h\boldsymbol{\Delta})} = \overline{c_j}\mathcal{C}^{-j}(h\boldsymbol{\Delta})$. Therefore, using $2\mathrm{Re}\{z\} = z + \overline{z}$, any Cayley filter (4) can be written as a conjugate-even Laurent polynomial w.r.t. $\mathcal{C}(h\boldsymbol{\Delta})$,

$$\mathbf{G} = c_0\mathbf{I} + \sum_{j=1}^r c_j\mathcal{C}^j(h\boldsymbol{\Delta}) + \overline{c_j}\mathcal{C}^{-j}(h\boldsymbol{\Delta}). \tag{5}$$

Since the spectrum of $\mathcal{C}(h\boldsymbol{\Delta})$ is in $e^{i\mathbb{R}}$, the operator $\mathcal{C}^j(h\boldsymbol{\Delta})$ can be thought of as a multiplication by a pure harmonic in the frequency domain $e^{i\mathbb{R}}$ for any integer power $j$,

$$\mathcal{C}^j(h\boldsymbol{\Delta}) = \boldsymbol{\Phi}\mathrm{diag}\big(\big[\mathcal{C}(h\lambda_1)\big]^j, \ldots, \big[\mathcal{C}(h\lambda_n)\big]^j\big)\boldsymbol{\Phi}^\top.$$

A Cayley filter can be thus seen as a multiplication by a finite Fourier expansions in the frequency domain $e^{i\mathbb{R}}$. Since (5) is conjugate-even, it is a (real-valued) trigonometric polynomial.

Note that any spectral filter can be formulated as a Cayley filter. Indeed, spectral filters $g(\boldsymbol{\Delta})$ are specified by the finite sequence of values $g(\lambda_1), \ldots, g(\lambda_n)$, which can be interpolated by a trigonometric polynomial. Moreover, since trigonometric polynomials are smooth, we expect low order Cayley filters to be well localized in some sense on the graph, as discussed later.

Finally, in definition (4) we use complex coefficients. If $c_j \in \mathbb{R}$ then (5) is an even cosine polynomial, and if $c_j \in i\mathbb{R}$ then (5) is an odd sine polynomial. Since the spectrum of $h\boldsymbol{\Delta}$ is in $\mathbb{R}_+$, it is mapped to the lower half-circle by $\mathcal{C}$, on which both cosine and sine polynomials are complete and can represent any spectral filter. However, it is beneficial to use general complex coefficients, since complex Fourier expansions are overcomplete in the lower half-circle, thus describing a larger variety of spectral filters of the same order without increasing the computational complexity of the filter.

**Spectral zoom.** To understand the essential role of the parameter $h$ in the Cayley filter, consider $\mathcal{C}(h\boldsymbol{\Delta})$. Multiplying $\boldsymbol{\Delta}$ by $h$ dilates its spectrum, and applying $\mathcal{C}$ on the result maps the non-negative spectrum to the complex half-circle. The greater $h$ is, the more the spectrum of $h\boldsymbol{\Delta}$ is spread apart in $\mathbb{R}_+$, resulting in better spacing of the smaller eigenvalues of $\mathcal{C}(h\boldsymbol{\Delta})$. On the other hand, the smaller $h$ is, the further away the high frequencies of $h\boldsymbol{\Delta}$ are from $\infty$, the better spread apart are the high frequencies of $\mathcal{C}(h\boldsymbol{\Delta})$ in $e^{i\mathbb{R}}$ (see Figure 1). Tuning the parameter $h$ allows thus to 'zoom' in to different parts of the spectrum, resulting in filters specialized in different frequency bands.

**Numerical properties.** The numerical core of the Cayley filter is the computation of $\mathcal{C}^j(h\boldsymbol{\Delta})\mathbf{f}$ for $j = 1, \ldots, r$, performed in a sequential manner. Let $\mathbf{y}_0, \ldots, \mathbf{y}_r$ denote the solutions of the following

linear recursive system,

$$\mathbf{y}_0 = \mathbf{f}, \qquad (h\boldsymbol{\Delta} + i\mathbf{I})\mathbf{y}_j = (h\boldsymbol{\Delta} - i\mathbf{I})\mathbf{y}_{j-1} \ , \ j = 1, \ldots, r. \tag{6}$$

Note that sequentially approximating $\mathbf{y}_j$ in (6) using the approximation of $\mathbf{y}_{j-1}$ in the rhs is stable, since $\mathcal{C}(h\boldsymbol{\Delta})$ is unitary and thus has condition number 1.

Equations (6) can be solved with matrix inversion exactly, but it costs $\mathcal{O}(n^3)$. An alternative is to use the Jacobi method,[1] which provides approximate solutions $\tilde{\mathbf{y}}_j \approx \mathbf{y}_j$. Let $\mathbf{J} = -(\mathrm{Diag}(h\boldsymbol{\Delta} + i\mathbf{I}))^{-1}\mathrm{Off}(h\boldsymbol{\Delta} + i\mathbf{I})$ be the Jacobi iteration matrix associated with equation (6). For the unnormalized Laplacian, $\mathbf{J} = (h\mathbf{D} + i\mathbf{I})^{-1}h\mathbf{W}$. Jacobi iterations for approximating (6) for a given $j$ have the form

$$\tilde{\mathbf{y}}_j^{(k+1)} = \mathbf{J}\tilde{\mathbf{y}}_j^{(k)} + \mathbf{b}_j, \quad \mathbf{b}_j = (\mathrm{Diag}(h\boldsymbol{\Delta} + i\mathbf{I}))^{-1}(h\boldsymbol{\Delta} - i\mathbf{I})\tilde{\mathbf{y}}_{j-1}, \tag{7}$$

initialized with $\tilde{\mathbf{y}}_j^{(0)} = \mathbf{b}_j$ and terminated after $K$ iterations, yielding $\tilde{\mathbf{y}}_j = \tilde{\mathbf{y}}_j^{(K)}$. The application of the approximate Cayley filter is given by $\widetilde{\mathbf{Gf}} = \sum_{j=0}^{r} c_j \tilde{\mathbf{y}}_j \approx \mathbf{Gf}$, and takes $\mathcal{O}(rKn)$ operations under the previous assumption of a sparse Laplacian. The method can be improved by normalizing $\|\tilde{\mathbf{y}}_j\|_2 = \|\mathbf{f}\|_2$.

Next, we give an error bound for the approximate filter. For the unnormalized Laplacian, let $d = \max_j\{d_{j,j}\}$ and $\kappa = \|\mathbf{J}\|_\infty = \frac{hd}{\sqrt{h^2d^2+1}} < 1$. For the normalized Laplacian, we assume that $(h\boldsymbol{\Delta}_n + i\mathbf{I})$ is dominant diagonal, which gives $\kappa = \|\mathbf{J}\|_\infty < 1$.

**Proposition 1.** *Under the above assumptions, $\frac{\|\mathbf{Gf} - \widetilde{\mathbf{Gf}}\|_2}{\|\mathbf{f}\|_2} \leq M\kappa^K$, where $M = \sqrt{n}\sum_{j=1}^{r} j\,|c_j|$ in the general case, and $M = \sum_{j=1}^{r} j\,|c_j|$ if the graph is regular.*

Proposition 1 is pessimistic in the general case, while requires strong assumptions in the regular case. We find that in most real life situations the behavior is closer to the regular case. It also follows from Proposition 1 that smaller values of the spectral zoom $h$ result in faster convergence, giving this parameter an additional numerical role of accelerating convergence.

**Complexity**. In practice, an accurate inversion of $(h\boldsymbol{\Delta} + i\mathbf{I})$ is not required, since the approximate inverse is combined with learned coefficients, which "compensate", as necessary, for the inversion inaccuracy. In a CayleyNet for a fixed graph, we fix the number of Jacobi iterations. Since the convergence rate depends on $\kappa$, that depends on the graph, different graphs may need different numbers of iterations. The convergence rate also depends on $h$. Since there is a trade-off between the spectral zoom amount $h$, and the accuracy of the approximate inversion, and since $h$ is a learnable parameter, the training finds the right balance between the spectral zoom amount and the inversion accuracy. We study the computational complexity of our method, as the number of edges $n$ of the graph tends to infinity. For every constant of a graph, e.g $d, \kappa$, we add the subscript $n$, indicating the number of edges of the graph. For the unnormalized Laplacian, we assume that $d_n$ and $h_n$ are bounded, which gives $\kappa_n < a < 1$ for some $a$ independent of $n$. For the normalized Laplacian, we assume that $\kappa_n < a < 1$. By Theorem 1, fixing the number of Jacobi iterations $K$ and the order of the filter $r$, independently of $n$, keeps the Jacobi error controlled. As a result, the number of parameters is $O(1)$, and for a Laplacian modeled as a sparse matrix, applying a Cayley filter on a signal takes $O(n)$ operations.

**Localization.** Unlike Chebyshev filters that have the small $r$-hop support, Cayley filters are rational functions supported on the whole graph. However, it is still true that Cayley filters are well localized on the graph. Let $\mathbf{G}$ be a Cayley filter and $\boldsymbol{\delta}_m$ denote a delta-function on the graph, defined as one at vertex $m$ and zero elsewhere. We show that $\mathbf{G}\boldsymbol{\delta}_m$ decays fast, in the following sense:

**Definition 2** (Exponential decay on graphs)**.** *Let $\mathbf{f}$ be a signal on the vertices of graph $\mathcal{G}$, $1 \leq p \leq \infty$, and $0 < \epsilon < 1$. Denote by $S \subseteq \{1, \ldots, n\}$ a subset of the vertices and by $S^c$ its complement. We say that the $L_p$-mass of $\mathbf{f}$ is supported in $S$ up to $\epsilon$ if $\|\mathbf{f}|_{S^c}\|_p \leq \epsilon \|\mathbf{f}\|_p$, where $\mathbf{f}|_{S^c} = (f_l)_{l \in S^c}$ is the restriction of $\mathbf{f}$ to $S^c$. We say that $\mathbf{f}$ has (graph) exponential decay about vertex $m$, if there exists some $\gamma \in (0,1)$ and $c > 0$ such that for any $k$, the $L_p$-mass of $\mathbf{f}$ is supported in $\mathcal{N}_{k,m}$ up to $c\gamma^k$. Here, $\mathcal{N}_{k,m}$ is the $k$-hop neighborhood of $m$.*

---

[1]We remind that the Jacobi method for solving $\mathbf{Ax} = \mathbf{b}$ consists in decomposing $\mathbf{A} = \mathrm{Diag}(\mathbf{A}) + \mathrm{Off}(\mathbf{A})$ and obtaining the solution iteratively as $\mathbf{x}^{(k+1)} = -(\mathrm{Diag}(\mathbf{A}))^{-1}\mathrm{Off}(\mathbf{A})\mathbf{x}^{(k)} + (\mathrm{Diag}(\mathbf{A}))^{-1}\mathbf{b}$.

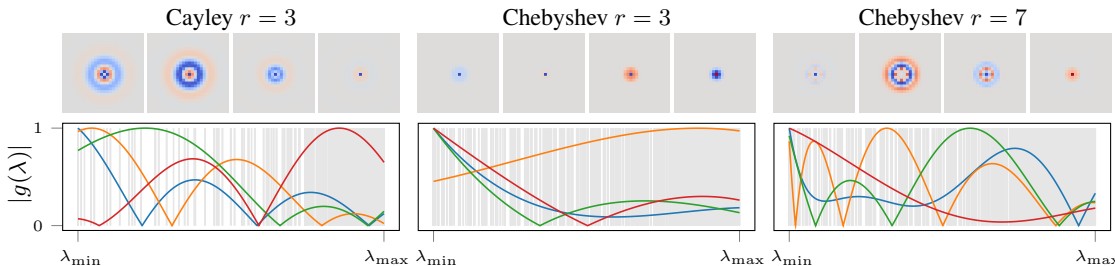

Figure 2: Filters (spatial domain, top and spectral domain, bottom) learned by CayleyNet (left) and ChebNet (center, right) on the MNIST dataset. Cayley filters are able to realize larger supports for the same order $r$.

**Remark 3.** *Note that Definition 2 is analogous to classical exponential decay on Euclidean space: $|f(x)| \leq R\gamma^{-x}$ iff for every ball $B_\rho$ of radius $\rho$ about 0, $\||f|_{B_\rho^c}\|_\infty \leq c\gamma^{-\rho} \|f\|_\infty$ with $c = \frac{R}{\|f\|_\infty}$.*

**Theorem 4.** *Let $\mathbf{G}$ be a Cayley filter of order $r$. Then, $\mathbf{G}\boldsymbol{\delta}_m$ has exponential decay about $m$ in $L_2$, with constants $c = 2M\frac{1}{\|\mathbf{G}\boldsymbol{\delta}_m\|_2}$ and $\gamma = \kappa^{1/r}$ (where $M$ and $\kappa$ are from Proposition 1).*

**Cayley vs Chebyshev.** Below, we compare the two classes of filters:

*Chebyshev as a special case of Cayley.* For a regular graph with $\mathbf{D} = d\mathbf{I}$, using Jacobi inversion based on zero iterations, we get that any Cayley filter of order $r$ is a polynomial of $\boldsymbol{\Delta}$ in the monomial base $\left(\frac{h\boldsymbol{\Delta}-i}{hd+i}\right)^j$. In this situation, a Chebyshev filter, which is a real valued polynomial of $\boldsymbol{\Delta}$, is a special case of a Cayley filter.

*Spectral zoom and stability.* Generally, both Chebyshev polynomials and trigonometric polynomials give stable approximations, optimal for smooth functions. However, this crude statement is over-simplified. One of the drawbacks in Chebyshev filters is the fact that the spectrum of $\boldsymbol{\Delta}$ is always mapped to $[-1, 1]$ in a linear manner, making it hard to specialize in small frequency bands. In Cayley filters, this problem is mitigated with the help of the spectral zoom parameter $h$. As an example, consider the community detection problem discussed in the next section. A graph with strong communities has a cluster of small eigenvalues near zero. Ideal filters $g(\boldsymbol{\Delta})$ for extracting the community information should be able to focus on this band of frequencies. Approximating such filters with Cayley polynomials, we zoom in to the band of interest by choosing the right $h$, and then project $g$ onto the space of trigonometric polynomials of order $r$, getting a good and stable approximation (Figure 3, bottom right). However, if we project $g$ onto the space of Chebyshev polynomials of order $r$, the interesting part of $g$ concentrated on a small band is smoothed out and lost (Figure 3, middle right). Thus, projections are not the right way to approximate such filters, and the stability of orthogonal polynomials cannot be invoked. When approximating $g$ on the small band using polynomials, the approximation will be unstable away from this band; small perturbations in $g$ will result in big perturbations in the Chebyshev filter away from the band. For this reason, we say that Cayley filters are more stable than Chebyshev filters.

*Regularity.* We found that in practice, low-order Cayley filters are able to model both very concentrated impulse-like filters, and wider Gabor-like filters. Cayley filters are able to achieve a wider range of filter supports with less coefficients than Chebyshev filters (Figure 2), making the Cayley class more regular than Chebyshev.

*Complexity.* Under the assumption of sparse Laplacians, both Cayley and Chebyshev filters incur linear complexity $\mathcal{O}(n)$. Besides, the new filters are equally simple to implement as Chebyshev filters; as seen in Eq.7, they boil down to simple sparse matrix-vector multiplications providing a GPU friendly implementation.

## 4 RESULTS

**Experimental settings.** We test the proposed CayleyNets reproducing the experiments of Defferrard et al. (2016); Kipf & Welling (2016); Monti et al. (2017a;a) and using ChebNet (Defferrard et al. (2016)) as our main baseline method. All the methods were implemented in TensorFlow of M. Abadi et. al. (2016). The experiments were executed on a machine with a 3.5GHz Intel Core i7 CPU, 64GB of RAM, and NVIDIA Titan X GPU with 12GB of RAM. SGD+Momentum and Adam

(Kingma & Ba (2014)) optimization methods were used to train the models in MNIST and the rest of the experiments, respectively. Training and testing were always done on disjoint sets.

**Community detection.**    We start with an experiment on a synthetic graph consisting of 15 communities with strong connectivity within each community and sparse connectivity across communities (Figure 3, left). Though rather simple, such a dataset allows to study the behavior of different algorithms in controlled settings. On this graph, we generate noisy step signals, defined as $f_i = 1 + \sigma_i$ if $i$ belongs to the community, and $f_i = \sigma_i$ otherwise, where $\sigma_i \sim \mathcal{N}(0, 0.3)$ is Gaussian i.i.d. noise. The goal is to classify each such signal according to the community it belongs to. The neural network architecture used for this task consisted of a spectral convolutional layer (based on Chebyshev or Cayley filters) with 32 output features, a mean pooling layer, and a softmax classifier for producing the final classification into one of the 15 classes. The classification accuracy is shown in Figure 3 (right, top) along with examples of learned filters (right, bottom). We observe that CayleyNet significantly outperforms ChebNet for smaller filter orders, with an improvement as large as 80%. Studying the filter responses, we note that due to the capability to learn the spectral zoom parameter, CayleyNet allows to generate band-pass filters in the low-frequency band that discriminate well the communities ( Figure 3 bottom right).

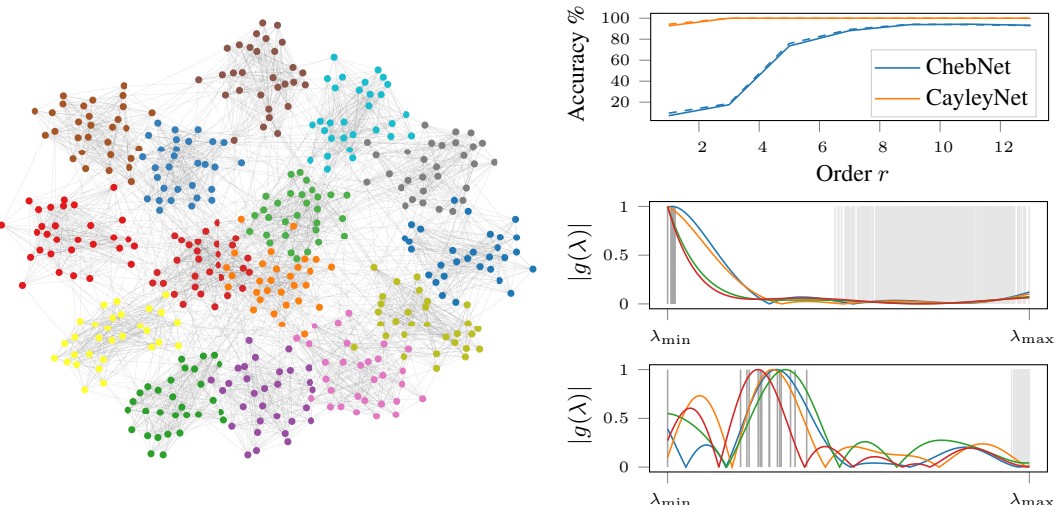

Figure 3: Left: synthetic 15-communities graph. Right: community detection accuracy of ChebNet and CayleyNet (top); normalized responses of four different filters learned by ChebNet (middle) and CayleyNet (bottom). Grey vertical lines represent the frequencies of the normalized Laplacian ($\tilde{\lambda} = 2\lambda_n^{-1}\lambda - 1$ for ChebNet and $C(\lambda) = (h\lambda - i)/(h\lambda + i)$ unrolled to a real line for CayleyNet). Note how thanks to spectral zoom property Cayley filters can focus on the band of low frequencies (dark grey lines) containing most of the information about communities.

**Complexity.**    We experimentally validated the computational complexity of our model applying filters of different order $r$ to synthetic 15-community graphs of different size $n$ using exact matrix inversion and approximation with different number of Jacobi iterations (Figure 4 center and right, Figure 6 in the appendix). All times have been computed running 30 times the considered models and averaging the final results. As expected, approximate inversion guarantees $\mathcal{O}(n)$ complexity. We further conclude that typically very few Jacobi iterations are required (Figure 4, left shows that our model with just one Jacobi iteration outperforms ChebNet for low-order filters on the community detection problem).

**MNIST.**    Following Defferrard et al. (2016); Monti et al. (2017a), for a toy example, we approached the classical MNIST digits classification as a learning problem on graphs. Each pixel of an image is a vertex of a graph (regular grid with 8-neighbor connectivity), and pixel color is a signal on the graph. We used a graph CNN architecture with two spectral convolutional layers based on Chebyshev and Cayley filters (producing 32 and 64 output features, respectively), interleaved with pooling layers performing 4-times graph coarsening using the Graclus algorithm (Dhillon et al. (2007)), and

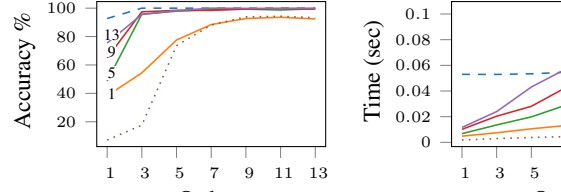 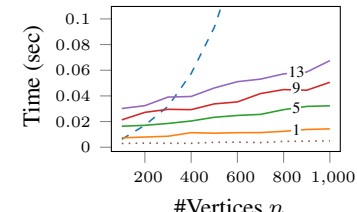

Figure 4: Left: community detection test accuracy as function of filter order $r$. Center and right: computational complexity (test times on batches of 100 samples) as function of filter order $r$ and graph size $n$. Shown are exact matrix inversion (dashed) and approximate Jacobi with different number of iterations (colored). For reference, ChebNet is shown (dotted).

finally a fully-connected layer (this architecture replicates the classical LeNet5, LeCun et al. (1998), architecture, which is shown for comparison). MNIST classification results are reported in Table 1. CayleyNet (11 Jacobi iterations) achieves the same (near perfect) accuracy as ChebNet with filters of lower order ($r = 12$ vs 25).Examples of filters learned by ChebNet and CayleyNet are shown in Figure 2. 0.1776 +/- 0.06079 sec and 0.0268 +/- 0.00841 sec are respectively required by CayleyNet and ChebNet for analyzing a batch of 100 images at test time.

Table 1: Test accuracy obtained with different methods on the MNIST dataset.

| Model | Order | Accuracy | #Params |
|---|---|---|---|
| LeNet5 | - | 99.33% | 1.66M |
| ChebNet | 25 | 99.14% | 1.66M |
| CayleyNet | 12 | 99.18% | 1.66M |

Table 2: Test accuracy of different methods on the CORA dataset. Scaled Laplacian and normalized Laplacian with real polynomials of order 1 have been respectively exploited for CayleyNet and ChebNet.

| Method | Accuracy | #Params |
|---|---|---|
| DCNN (Atwood & Towsley (2016)) | 86.01 % | 47K |
| GCN (Kipf & Welling (2016)) | 86.64 % | 47K |
| ChebNet (Defferrard et al. (2016)) | 87.07 % | 46K |
| **CayleyNet** | **88.09 %** | 46K |

**Citation network.** Next, we address the problem of vertex classification on graphs using the popular CORA citation graph, Sen et al. (2008). Each of the 2708 vertices of the CORA graph represents a scientific paper, and an undirected unweighted edge represents a citation (5429 edges in total). For each vertex, a 1433-dimensional binary feature vector representing the content of the paper is given. The task is to classify each vertex into one of the 7 groundtruth classes. We split the graph into training (1,708 vertices), validation (500 vertices) and test (500 vertices) sets, for simulating the labeled and unlabeled information. We train ChebNet and CayleyNet with the architecture presented in Kipf & Welling (2016); Monti et al. (2017a) (two spectral convolutional layers with 16 and 7 outputs), DCNN (Atwood & Towsley (2016)) with 2 diffusion layer (10 hidden features and 2 diffusion hops) and GCN (Kipf & Welling (2016)) with 3 convolutional layer (32 and 16 hidden features). Figure 5 compares ChebNets and CayleyNets, in a number of different settings. Since ChebNets require Laplacians with spectra bounded in $[-1, 1]$, we consider both the normalized Laplacian (the two left figures), and the scaled unnormalized Laplacian ($2\mathbf{\Delta}/\lambda_{max} - \mathbf{I}$), where $\mathbf{\Delta}$ is the unnormalized Laplacian and $\lambda_{max}$ is its largest eigenvalue (the two right figures). For fair comparison, we fix the order of the filters (top figures), and fix the overall number of network parameters (bottom figures). In the bottom figure, the Cayley filters are restricted to even cosine polynomials by considering only real filter coefficients. Table 2 shows a comparison of the performance obtained with different methods (all architectures roughly present the same amount of parameters). The best CayleyNets consistently outperform the best competitors.

**Recommender system.** In our final experiment, we applied CayleyNet to recommendation system, formulated as matrix completion problem on user and item graphs, Monti et al. (2017a). The task is, given a sparsely sampled matrix of scores assigned by users (columns) to items (rows), to fill in the missing scores. The similarities between users and items are given in the form of column and row graphs, respectively. Monti et al. (2017a) approached this problem as learning with a Recurrent Graph CNN (RGCNN) architecture, using an extension of ChebNets to matrices defined on multiple graphs in order to extract spatial features from the score matrix; these features are then fed into an

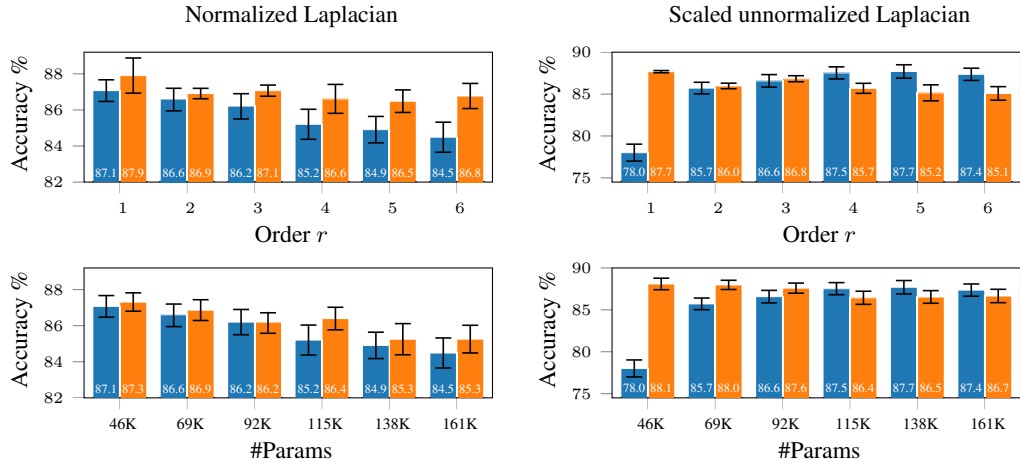

Figure 5: ChebNet (blue) and CayleyNet (orange) test accuracies obtained on the CORA dataset for different polynomial orders. Polynomials with complex coefficients (top) and real coefficients (bottom) have been exploited with CayleyNet in the two analysis. Orders 1 to 6 have been used in both comparisons. The best CayleyNet consistently outperform the best ChebNet requiring at the same time less parameters (CayleyNet with order $r$ and complex coefficients requires a number of parameters equal to ChebNet with order $2r$).

RNN producing a sequential estimation of the missing scores. Here, we repeated verbatim their experiment on the MovieLens dataset (Miller et al. (2003)), replacing Chebyshev filters with Cayley filters. We used separable RGCNN architecture with two CayleyNets of order $r = 4$ employing 15 Jacobi iterations. The results are reported in Table 3. To present a complete comparison we further extended the experiments reported in Monti et al. (2017a) by training sRGCNN with ChebNets of order 8, this provides an architecture with same number of parameters as the exploited CayleyNet (23K coefficients). Our version of sRGCNN outperforms all the competing methods, including the previous result with Chebyshev filters reported in Monti et al. (2017a). sRGCNNs with Chebyshev polynomials of order 4 and 8 respectively require 0.0698 +/- 0.00275 sec and 0.0877 +/- 0.00362 sec at test time, sRGCNN with Cayley polynomials of order 4 and 15 jacobi iterations requires 0.165 +/- 0.00332 sec.

Table 3: Performance (RMSE) of different matrix completion methods on the MovieLens dataset.

| Method | RMSE |
| --- | --- |
| MC (Candes & Recht (2012)) | 0.973 |
| IMC (Jain & Dhillon (2013); Xu et al. (2013)) | 1.653 |
| GMC (Kalofolias et al. (2014)) | 0.996 |
| GRALS (Rao et al. (2015)) | 0.945 |
| sRGCNN$_{Cheby,r=4}$ (Monti et al. (2017a)) | 0.929 |
| sRGCNN$_{Cheby,r=8}$ (Monti et al. (2017a)) | 0.925 |
| **sRGCNN$_{Cayley}$** | **0.922** |

## 5 CONCLUSIONS

In this paper, we introduced a new efficient spectral graph CNN architecture that scales linearly with the dimension of the input data. Our architecture is based on a new class of complex rational Cayley filters that are localized in space, can represent any smooth spectral transfer function, and are highly regular. The key property of our model is its ability to specialize in narrow frequency bands with a small number of filter parameters, while still preserving locality in the spatial domain. We validated these theoretical properties experimentally, demonstrating the superior performance of our model in a broad range of graph learning problems.

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

APPENDIX

PROOF OF PROPOSITION 1

First note the following classical result for the approximation of $\mathbf{A}\mathbf{x} = \mathbf{b}$ using the Jacobi method: if the initial condition is $\mathbf{x}^{(0)} = \mathbf{0}$, then $(\mathbf{x} - \mathbf{x}^{(k)}) = \mathbf{J}^k\mathbf{x}$. In our case, note that if we start with initial condition $\tilde{\mathbf{y}}_j^{(0)} = 0$, the next iteration gives $\tilde{\mathbf{y}}_j^{(0)} = \mathbf{b}_j$, which is the initial condition from our construction. Therefore, since we are approximating $\mathbf{y}_j = \mathcal{C}(h\boldsymbol{\Delta})\tilde{\mathbf{y}}_{j-1}$ by $\tilde{\mathbf{y}}_j = \tilde{\mathbf{y}}_j^{(K)}$, we have

$$\mathcal{C}(h\boldsymbol{\Delta})\tilde{\mathbf{y}}_{j-1} - \tilde{\mathbf{y}}_j = \mathbf{J}^{K+1}\mathcal{C}(h\boldsymbol{\Delta})\tilde{\mathbf{y}}_{j-1} \tag{8}$$

Define the approximation error in $\mathcal{C}(h\boldsymbol{\Delta})^j\mathbf{f}$ by

$$e_j = \frac{\left\|\mathcal{C}^j(h\boldsymbol{\Delta})\mathbf{f} - \tilde{\mathbf{y}}_j\right\|_2}{\left\|\mathcal{C}^j(h\boldsymbol{\Delta})\mathbf{f}\right\|_2}.$$

By the triangle inequality, by the fact that $\mathcal{C}^j(h\boldsymbol{\Delta})$ is unitary, and by (8)

$$\begin{aligned}
e_j &\leq \frac{\left\|\mathcal{C}^j(h\boldsymbol{\Delta})\mathbf{f} - \mathcal{C}(h\boldsymbol{\Delta})\tilde{\mathbf{y}}_{j-1}\right\|_2}{\left\|\mathcal{C}^j(h\boldsymbol{\Delta})\mathbf{f}\right\|_2} + \frac{\left\|\mathcal{C}(h\boldsymbol{\Delta})\tilde{\mathbf{y}}_{j-1} - \tilde{\mathbf{y}}_j\right\|_2}{\left\|\mathcal{C}^j(h\boldsymbol{\Delta})\mathbf{f}\right\|_2} \\
&= \frac{\left\|\mathcal{C}^{j-1}(h\boldsymbol{\Delta})\mathbf{f} - \tilde{\mathbf{y}}_{j-1}\right\|_2}{\left\|\mathcal{C}^{j-1}(h\boldsymbol{\Delta})\mathbf{f}\right\|_2} + \frac{\left\|\mathbf{J}^{K+1}\mathcal{C}(h\boldsymbol{\Delta})\tilde{\mathbf{y}}_{j-1}\right\|_2}{\left\|\mathbf{f}\right\|_2} \\
&\leq e_{j-1} + \left\|\mathbf{J}^{K+1}\right\|_2 \frac{\left\|\mathcal{C}(h\boldsymbol{\Delta})\tilde{\mathbf{y}}_{j-1}\right\|_2}{\left\|\mathbf{f}\right\|_2} = e_{j-1} + \left\|\mathbf{J}^{K+1}\right\|_2 \frac{\left\|\tilde{\mathbf{y}}_{j-1}\right\|_2}{\left\|\mathbf{f}\right\|_2} \\
&\leq e_{j-1} + \left\|\mathbf{J}^{K+1}\right\|_2 (1 + e_{j-1})
\end{aligned} \tag{9}$$

where the last inequality is due to

$$\left\|\tilde{\mathbf{y}}_{j-1}\right\|_2 \leq \left\|\mathcal{C}^{j-1}(h\boldsymbol{\Delta})\mathbf{f}\right\|_2 + \left\|\mathcal{C}^{j-1}(h\boldsymbol{\Delta})\mathbf{f} - \tilde{\mathbf{y}}_{j-1}\right\|_2 = \left\|\mathbf{f}\right\|_2 + \left\|\mathbf{f}\right\|_2 e_{j-1}.$$

Now, using standard norm bounds, in the general case we have $\left\|\mathbf{J}^{K+1}\right\|_2 \leq \sqrt{n}\left\|\mathbf{J}^{K+1}\right\|_\infty$. Thus, by $\kappa = \left\|\mathbf{J}\right\|_\infty$ we have

$$e_j \leq e_{j-1} + \sqrt{n}\left\|\mathbf{J}\right\|_\infty^{K+1}(1 + e_{j-1}) = (1 + \sqrt{n}\kappa^{K+1})e_{j-1} + \sqrt{n}\kappa^{K+1}.$$

The solution of this recurrent sequence is

$$e_j \leq (1 + \sqrt{n}\kappa^{K+1})^j - 1 = j\sqrt{n}\kappa^{K+1} + O(\kappa^{2K+2}).$$

If we use the version of the algorithm, in which each $\tilde{\mathbf{y}}_j$ is normalized, we get by (9) $e_j \leq e_{j-1} + \sqrt{n}\kappa^{K+1}$. The solution of this recurrent sequence is

$$e_j \leq j\sqrt{n}\kappa^{K+1}.$$

We denote in this case $M_j = j\sqrt{n}$

In case the graph is regular, we have $\mathbf{D} = d\mathbf{I}$. In the non-normalized Laplacian case,

$$\mathbf{J} = -(hd\mathbf{I} + i\mathbf{I})^{-1}h(\boldsymbol{\Delta} - d\mathbf{I}) = \frac{h}{hd + i}(d\mathbf{I} - \boldsymbol{\Delta}) = \frac{h}{hd + i}\mathbf{W}. \tag{10}$$

The spectral radius of $\boldsymbol{\Delta}$ is bounded by $2d$. This can be shown as follows. a value $\lambda$ is not an eigenvalue of $\boldsymbol{\Delta}$ (namely it is a regular value) if and only if $(\boldsymbol{\Delta} - \lambda\mathbf{I})$ is invertible. Moreover, the matrix $(\boldsymbol{\Delta} - \lambda\mathbf{I})$ is strictly dominant diagonal for any $|\lambda| > 2d$. By Levy–Desplanques theorem, any strictly dominant diagonal matrix is invertible, which means that all of the eigenvalues of $\boldsymbol{\Delta}$ are less than $2d$ in their absolute value. As a result, the spectral radius of $(d\mathbf{I} - \boldsymbol{\Delta})$ is realized on the smallest eigenvalue of $\boldsymbol{\Delta}$, namely it is $|d - 0| = d$. This means that the specral radius of $\mathbf{J}$ is $\frac{hd}{\sqrt{h^2d^2+1}}$. As a result $\left\|\mathbf{J}\right\|_2 = \frac{hd}{\sqrt{h^2d^2+1}} = \kappa$. We can now continue from (9) to get

$$e_j \leq e_{j-1} + \left\|\mathbf{J}\right\|_2^{K+1}(1 + e_{j-1}) = e_{j-1} + \kappa^{K+1}(1 + e_{j-1}).$$

As before, we get $e_j \leq j\kappa^{K+1} + O(\kappa^{2K+2})$, and $e_j \leq j\kappa^{K+1}$ if each $\tilde{\mathbf{y}}_j$ is normalized. We denote in this case $M_j = j$.

In the case of the normalized Laplacian of a regular graph, the spectral radius of $\boldsymbol{\Delta}_n$ is bounded by 2, and the diagonal entries are all 1. Equation (10) in this case reads $\mathbf{J} = \frac{h}{h+i}(\mathbf{I} - \boldsymbol{\Delta}_n)$, and $\mathbf{J}$ has spectral radius $\frac{h}{\sqrt{h^2+1}}$. Thus $\|\mathbf{J}\|_2 = \frac{h}{\sqrt{h^2+1}} = \kappa$ and we continue as before to get $e_j \leq j\kappa^{K+1}$ and $M_j = j$.

In all cases, we have by the triangle inequality

$$\frac{\left\|\mathbf{Gf} - \widetilde{\mathbf{Gf}}\right\|_2}{\|\mathbf{f}\|_2} \leq \sum_{j=1}^{r} |c_j| \frac{\left\|\mathcal{C}^j(h\boldsymbol{\Delta})\mathbf{f} - \tilde{\mathbf{y}}_j\right\|_2}{\left\|\mathcal{C}^j(h\boldsymbol{\Delta})\mathbf{f}\right\|_2} = \sum_{j=1}^{r} |c_j|\, e_j \leq \sum_{j=1}^{r} M_j\, |c_j|\, \kappa^{K+1}.$$

PROOF OF THEOREM 4

In this proof we approximate $\mathbf{G}\boldsymbol{\delta}_m$ by $\widetilde{\mathbf{G}\boldsymbol{\delta}}_m$. Note that the signal $\boldsymbol{\delta}_m$ is supported on one vertex, and in the calculation of $\widetilde{\mathbf{G}\boldsymbol{\delta}}_m$, each Jacobi iteration increases the support of the signal by 1-hop. Therefore, the support of $\widetilde{\mathbf{G}\boldsymbol{\delta}}_m$ is the $r(K+1)$-hop neighborhood $\mathcal{N}_{r(K+1),m}$ of $m$. Denoting $l = r(K+1)$, and using Proposition 1, we get

$$
\begin{aligned}
\left\|\mathbf{G}\boldsymbol{\delta}_m - \mathbf{G}\boldsymbol{\delta}_m|_{\mathcal{N}_{l,m}}\right\|_2 &\leq \left\|\mathbf{G}\boldsymbol{\delta}_m - \widetilde{\mathbf{G}\boldsymbol{\delta}}_m\right\|_2 + \left\|\widetilde{\mathbf{G}\boldsymbol{\delta}}_m - \mathbf{G}\boldsymbol{\delta}_m|_{\mathcal{N}_{l,m}}\right\|_2 \\
&\leq \left\|\mathbf{G}\boldsymbol{\delta}_m - \widetilde{\mathbf{G}\boldsymbol{\delta}}_m\right\|_2 + \left\|\widetilde{\mathbf{G}\boldsymbol{\delta}}_m - \mathbf{G}\boldsymbol{\delta}_m\right\|_2 \\
&= 2\left\|\mathbf{G}\boldsymbol{\delta}_m - \widetilde{\mathbf{G}\boldsymbol{\delta}}_m\right\|_2 \leq 2M\kappa^{K+1}\|\boldsymbol{\delta}_m\|_2 \\
&= 2M(\kappa^{1/r})^l.
\end{aligned}
\tag{11}
$$

COMPUTATIONAL COMPLEXITY

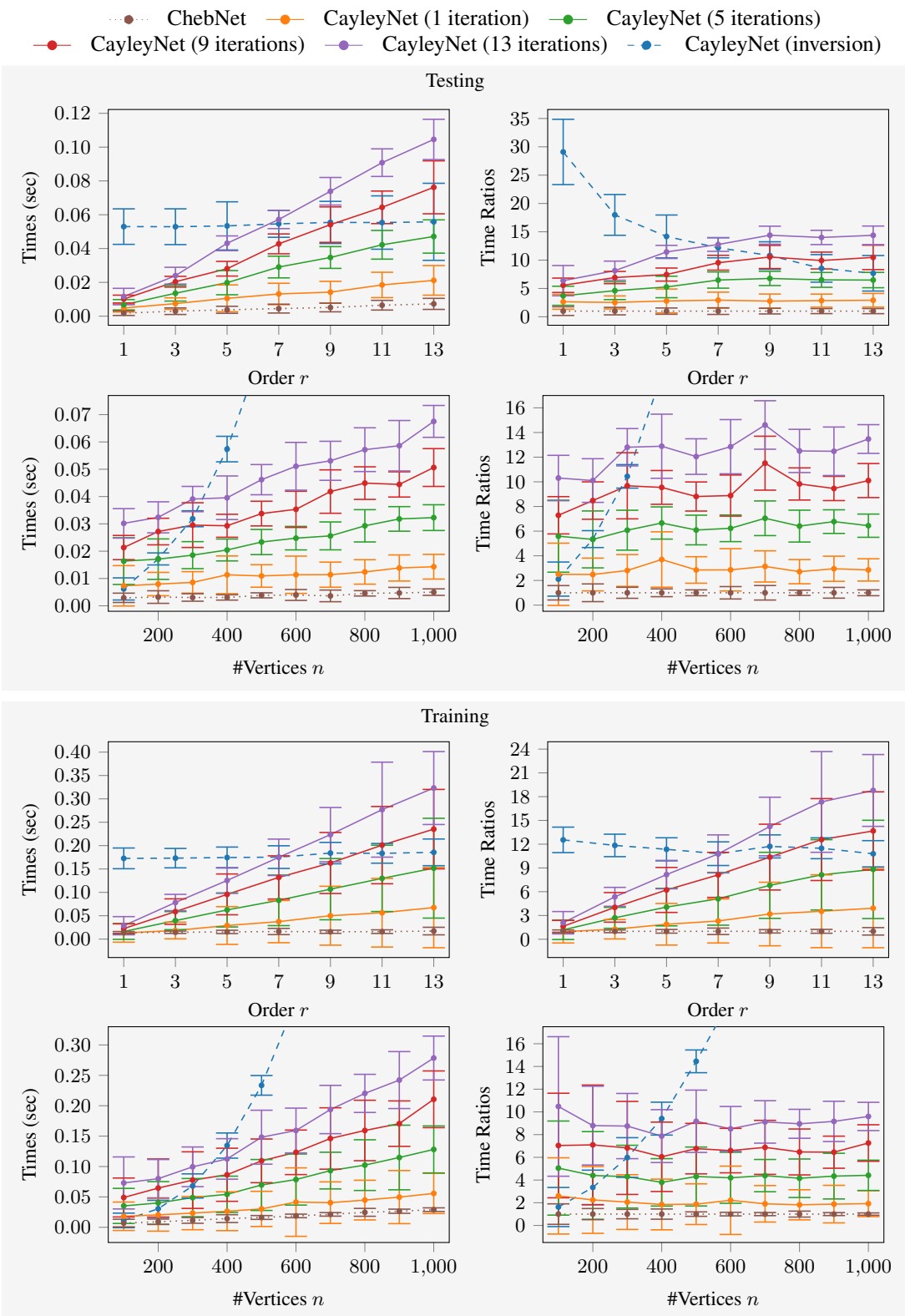

Figure 6: Test (above) and training (below) times with corresponding ratios as function of filter order $r$ and graph size $n$ on our community detection dataset.

