# OpenReview forum: "CAYLEYNETS: SPECTRAL GRAPH CNNS WITH COMPLEX RATIONAL FILTERS"
_ICLR.cc/2018/Conference — Reject_

### Official Review · AnonReviewer2 · 2017-11-26
**Interesting new filter for graph CNNs, although experiments are not fully convincing**

**Rating:** 6
**Confidence:** 3

**Review:**

The paper proposes a new filter for spectral analysis on graphs for graph CNNs. The filter is a rational function based on the Cayley transform. Unlike other popular variants, it is not strictly supported on a small graph neighborhood, but the paper proves an exponential-decay property on the norm of a filtered vertex indicator function.

The paper argues that Cayley filters allow better spectral localization than Chebyshev filters. While Chebyshev filters can be applied efficiently using a recursive method, evaluation of a  Cayley filter of order r requires solving r linear system in dimension corresponding to the number of vertices, which is expensive. The paper proposes to stop after a small number of iterations of Jacobi's method to alleviate this problem.

The paper is clear and well written.

The proposed method seems of interest, although I find the experimental section only partly convincing.

There seems to be a tradeoff here. The paper demonstrates that CayleyNet achieves similar efficiency as ChebNet in multiple experiments while using smaller filter orders. Although using smaller filter orders (and better-localized filters) is an interesting property, it is not necessarily a key objective, especially as this seems to come at the cost of a significantly increased computational complexity.

The paper could help us understand this tradeoff better. For instance:
- Middle and right panels of Figure 4 could use a more precise Y scale. How much slower is CayleyNet here with respect to the ChebNet?

- Figure 4 mentions time corresponds to "test times on batches of 100 samples". Is this an average value over multiple 100-sample batches? What is the standard deviation? How do the training times compare?

- MNIST accuracies are very similar (and near perfect) -- how did the training and testing time compare? Same for the MovieLens experiment. The improvement in performance is rather small, what is the corresponding computational cost?

- CORA results are a bit confusing to me. The filter orders used here are very small, and the best amongst the values considered seems to be r=1. Is there a reason only such small values have been considered? Is this a fair evaluation of ChebNet which may possibly perform better with larger filter orders?

- The paper could provide some insights as to why ChebNet is unable to work with unnormalized Laplacians while CayleyNet is (and why the ChebNet performance seems to get worse and worse as r increases?).

---

> ### Author Response · Authors · 2017-12-29
> **CORA/MNIST/MovieLens test and training times plus CayleyNet vs ChebNet on unnormalized Laplacians**
>
> We thank anonymous Reviewer2 for the work he/she provided. We present here various insights on the highlighted points.
>
> We first note that smaller filter orders are not the key objective. They lead to more regular filter spaces, which ultimately leads to less overfitting, and better accuracy, which is the key goal. This is evident in the experimental results.
>
>
> 1. Community dataset test/training times
>
> We agree with the reviewer that the scale proposed on the y axis of Fig. 4 is not detailed enough, which will be fixed in the revision. All test times have been computed running 30 times our models with batches of 100 samples and averaging times across batches (thus the reported times should be considered as mean test times per batch).
>
> In order to provide a better understanding, we attach here 2 anonymous links to figures showing ratios between times obtained with CayleyNet and ChebNet (i.e. test_time_CayleyNet / test_time_ChebNet):
>
> https://ibb.co/bD4xNG
> https://ibb.co/jbNdUw
>
> We will add these plots as supplementary material in our final revisions.
>
>
> Standard deviations have been avoided in our analysis since do not add much to what already presented in Fig.4. For completeness, we attach here 2 links showing mean test times and corresponding standard deviations:
>
> https://ibb.co/jnODwb
> https://ibb.co/nezWhG
>
> Finally, training times have been avoided in the paper for reasons of space. In general they present a similar trend to test times: https://ibb.co/gUjE2G, https://ibb.co/bHAnNG, https://ibb.co/kbrYwb, https://ibb.co/kcYP2G. We will add training times in the final version of this work.
>
>
> 2. CORA accuracies
>
> As also requested by Reviewer 3, we further extended our analysis with additional orders. The best CayleyNet still outperform the competitor requiring at the same time a smaller amount of parameters (see point 1 and 2 of our response to Reviewer 3).
>
>
> 3. MNIST/MovieLens performance and test times
>
> Performance obtained over the MNIST dataset have been computed by means of 11 Jacobi iterations. Test time required by the proposed approach thus appears equal to 0.1776 +/- 0.06079 sec wrt the 0.0268 +/- 0.00841 sec required by ChebNet per batch (batch size = 100 images). We stress that MNIST digit classification just represents a toy example for ensuring the performance of our approach on a well known benchmark in standard conditions and should not be considered as a valuable example for proving the superior capabilities of the proposed spectral filters.
>
> For what concern MovieLens, ChebNets with order 4 and 8 respectively require 0.0698 +/- 0.00275 sec and 0.0877 +/- 0.00362 sec at test time, CayleyNet with order 4 and 15 jacobi iterations requires instead 0.165 +/- 0.00332 sec. As presented to Reviewer 3, the only modest improvement obtained by CayleyNet on this dataset is due to the construction of the graph.
>
>
> 4. ChebNet and unnormalized Laplacians
>
> Chebyshev polynomials are only well defined in the interval [-1,-1], and plugging in values away from this interval leads an ill behaved system. Following the comments of Reviewer 3 we updated the paper to compare the two methods over the scaled version of the unnormalized laplace operator proposed by Defferrard et al.
> The eigenvalues of the unnormalized laplacian \Delta are bounded by max{d(u)+d(v):uv∈E} (where d(u) corresponds to the degree of node u and E is the edge set, doi.org/10.1080/03081088508817681). In ChebNet, Defferrard et al. proposed to divide the unnormalized laplace operator by the maximum eigenvalue (thus producing a contraction from the original laplacian). The Chebyshev polynomial basis is well defined on this normalized version of the Laplacian. However, a side effect of this normalization is that all of the ``macroscopic frequencies’’ get squeezed near zero, and thus Chebyshev polynomials cannot separate them. This phenomenon is avoided in CayleyNets, as explained in the “Cayley vs Chebyshev” section. In the updated comparison, CayleyNet still achieves better performance while requiring a lower amount of parameters.
>
> Regarding the last remark of the reviewer, the performance gets worse as the filter order increases due to overfitting.

---

### Official Review · AnonReviewer3 · 2017-11-28
**Propose new filters based on Cayley transform -- interesting filter, but unconvincing theory / experiments**

**Rating:** 4
**Confidence:** 3

**Review:**

Summary: This paper proposes a new graph-convolution architecture, based on Cayley transform of the matrix. Succinctly, if L denotes the Laplacian of a graph, this filter corresponds to an operator that is a low degree polynomial of C(L) = (hL - i)/(hL+i), where h is a scalar and i denotes sqrt(-1). The authors contend that such filters are interesting because they can 'zoom' into a part of the spectrum, depending on the choice of h, and that C(L) is always a rotation matrix with eigenvalues with magnitude 1. The authors propose to compute them using Jacobi iteration (using the diagonal as a preconditioner), and present experimental results.

Opinion: Though the Cayley filters seem to have interesting properties,  I find the authors theoretical and experimental justification insufficient to conclude that they offer sufficient advantage over existing methods. I list my major criticisms below:
1. The comparison to Chebyshev filters  (small degree polynomials in the Chebyshev basis) at several places is unconvincing. The results on CORA (Fig 5a) compare filters with the same order, though Cayley filters have twice the number of variables for the same order as Chebyshev filters. Similarly for Fig 1, order 3 Cayley should be compared to Order 6 Chebyshev (roughly).

2. Since Chebyshev polynomials blow up exponentially when applied to values larger than 1, applying Chebyshev filters to unnormalized Laplacians (Fig 5b) is an unfair comparison.

3. The authors basically apply Jacobi iteration (gradient descent using a diagonal preconditioner) to estimate the Cayley filters, and contend that a constant number of iterations of Jacobi are sufficient. This ignores the fact that their convergence rate scales quadratically in h and the max-degree of the graph. Moreover, this means that the Filter is effectively a low degree polynomial in (D^(-1)A)^K, where A is the adjacency matrix of the graph, and K is the number of Jacobi iterations. It's unclear how (or why) a choice of K might be good, or why does it make sense to throw away all powers of D^(-1)Af, even though we're computing all of them.
Also, note that this means a K-fold increase in the runtime for each evaluation of the network, compared to the Chebyshev filter.

Among the other experimental results, the synthetic results do clearly convey a significant advantage at least over Chebyshev filters with the same number of parameters. The CORA results (table 2) do convey a small but clear advantage. The MNIST result seems a tie, and the comparison for MovieLens doesn't make it obvious that the number of parameters is the same.

Overall, this leads me to conclude that the paper presents insufficient justification to conclude that Cayley filters offer a significant advantage over existing work.

---

> ### Author Response · Authors · 2017-12-20
> **ChebNet - CayleyNet, a deeper comparison**
>
> We thank anonymous Reviewer3 for thorough and insightful comments. We have run extensive experiments requested by the reviewer and provide these results as well as our detailed response to his/her main concerns below. We will revise the paper to address these issues and our responses to them.
>
>
> 1. Number of coefficients: We agree that, since Cayley filters use complex coefficients while Chebyshev filters use real coefficients, in principle complex coefficients should be counted as twice more parameters. There are two different ways to make the number of parameters fairly comparable: (i) compare Cayley filters of order r vs Chebyshev filters of order 2*r (as suggested by Reviewer3), or (ii) use real coefficients in Cayley filters (as we note in our paper on p.4, in paragraph preceding Fig 1).
>
> We produce these two comparisons below, using the Cora dataset with symmetric normalized Laplacian (#params = #real coefficients; 1 complex coefficient is counted as 2 parameters):
>
> (i) Cayley filter with complex coefficients, twice lower order than Chebyshev filter:
>
> ChebNet order r=2 (#params = 69136) - Accuracy = 86.607986 +/- 0.65477967 (reported in paper)
> ChebNet order r=4 (#params = 115216) - Accuracy = 85.203995 +/- 0.83185506
> ChebNet order r=6 (#params = 161296) - Accuracy = 84.487999 +/- 0.83249897
>
> Complex CayleyNet order r=1 (#params = 69136) - Accuracy = 87.9  +/- 0.97508276 (reported in paper)
> Complex CayleyNet order r=2 (#params = 115216) - Accuracy = 86.9 +/- 0.28902602 (reported in paper)
> Complex CayleyNet order r=3  (#params = 161296) - Accuracy = 87.1 +/- 0.30883133 (reported in paper)
>
>
> (ii) Cayley filter using real coefficients, same order as Chebyshev filter:
>
> Real CayleyNet order r=1 (#params = 46096) - Accuracy = 87.311989 +/- 0.50936872
> Real CayleyNet order r=2 (#params = 69136) - Accuracy = 86.863991 +/- 0.57611096
> Real CayleyNet order r=3 (#params = 92176) - Accuracy = 86.147995 +/- 0.56823856
> Real CayleyNet order r=4 (#params = 115216) - Accuracy = 86.395996 +/- 0.62544805
> Real CayleyNet order r=5 (#params = 138256) - Accuracy = 85.251991 +/- 0.86330509
> Real CayleyNet order r=6 (#params = 161296) - Accuracy = 85.255997 +/- 0.76737374
>
> In both cases (i) and (ii), CayleyNet outperforms ChebNet (higher accuracy) for the same number of parameters.
>
>
> Furthermore, for the MovieLens experiment, the CayleyNet outperforms ChebNet (lower RMS error) when using lower polynomial order:
>
> Complex CayleyNet order r=4 (#params = 23126) - RMSE = 0.922 (reported in paper)
> ChebNet order r=8 (#params = 23124) - RMSE = 0.925
>
>
> We will include these results and a more detailed discussion regarding a fair comparison of the number of parameters.
>
>
> 2. Normalized vs Unnormalized Laplacian: We agree that poor performance of ChebNet in case of unnormalized Laplacian can be attributed to large eigenvalues. As requested by Reviewer3, we reproduce this experiment using scaled unnormalized Laplacian (2*Delta/lambda_max - I) to ensure the magnitude of its eigenvalues is <= 1, thus avoiding the numerical instability raised by the reviewer. We note that in our approach, no such scaling is necessary, since the eigenvalues of any Laplacian are mapped to the complex unit circle and thus automatically numerically stable.
>
> The best performing models on Cora dataset with scaled unnormalized Laplacian are reported below:
>
> ChebNet order r=7 (#params = 184336) - Accuracy = 87.232002 +/- 0.68511164
> CayleyNet order r=1 (#params = 69136) - Accuracy = 87.676003 +/- 0.13199957
>
> Thus, CayleyNet outperforms ChebNet (accuracy of 87.68% vs 87.23%) at the same time requiring significantly less parameters (69K vs 184K). We will update the results reported in the paper for the unnormalized Laplacian by this experiment.

---

> > ### Author Response · Authors · 2017-12-20
> > **From a more theoretical point of view**
> >
> >
> > 3. Jacobi approximate inversion: We agree that the choice of parameters and their tradeoffs deserves a more detailed discussion. First of all, please note that for K=0, we obtain standard polynomials expressed in the basis (x-i)^j, which makes ChebNet a particular case of our method.
> >
> > For K>0, the iterations of the Jacobi method for matrix inversion can indeed be interpreted as a polynomial of degree K in (D^(-1)A)^K. Please note that the coefficients of this polynomial are fixed and not learnable, otherwise we would have too many parameters prone to overfitting.
> >
> > Most importantly, we argue that an accurate inversion of the matrix is not needed and thus use a fixed number K of Jacobi iterations. The reason is that the application of the approximate inverse to the input signal (\tilde{y}in our notation) is then combined with learned coefficients, which “compensate”, as necessary, for the inversion inaccuracy.
> > Such behavior is well-documented in the literature in other contexts of model compression and accelerated convergence of iterative algorithms (see e.g. Gregor&LeCun ICML 2010 and numerous follow-up works); for example, learning sparse signal coding by unrolling iterative shrinkage algorithms (FISTA) into a neural network, where each layer emulates an iteration of the original algorithm but has extra learnable parameters. It is shown that FISTA networks with just a few layers outperform hundreds or thousands of iterations of the original algorithm thanks to the learnable parameters. We believe that a more careful analysis of this phenomenon is an interesting future work direction.
> >
> > Reviewer 3 rightfully noted that the convergence rate of the Jacobi inversion depends on h. Indeed, there is a trade-off between the value of h, and the accuracy of the approximate inversion. Since h is a learnable parameter, ultimately, the training finds the right balance between the spectral zoom amount and the inversion accuracy. Moreover, as Reviewer 3 noted, the accuracy of the Jacobi inversion also depends on the max degree of the graph. This means that different graphs may require different h and number K of Jacobi iterations. However, once the graph is fixed, the max degree is fixed, so the number of iterations corresponding to the graph is also fixed. Naturally, different problems based on different graphs, require different numbers of iterations. We do not ignore this fact, but on the contrary, report it in Proposition 1. The importance of this proposition is to set a uniform bound on the convergence rate, that only depends on the graph. As a result, the number of iterations can be globally fixed for each graph, while as noted above, the training of h is underlied by a trade-off between accuracy and spectral zoom. We will make these facts more explicit in the paper.
> >
> > 4. Experimental results: As shown by our toy experiment (communities graph), the advantage of our method is especially pronounced when the spectrum of the graph Laplacian has clustered eigenvalues (in particular, this is the case of graphs with strong communities, where there are multiple near-zero eigenvalues). The non-linear transformation of the eigenvalues by means of the Cayley transform and the spectral zoom property allow to achieve filters that better separate these frequencies. We thus expect our method to be especially advantageous in the analysis of social networks where strong communities are typically observed.
> >
> > The citation network Cora is well known to have strong community structure, hence the pronounced advantage of our method.
> >
> > The fact that experiments on MNIST does not show a significant advantage of CayleyNet is that being planar regular graph (2D grid), there is no clustering of eigenvalues. We regard MNIST as a mere “sanity check”, to ensure that in the simple Euclidean setting our approach is as good as classical CNN (LeNet).
> >
> > A slightly different situations appear in the MovieLens experiments. While we would typically expect similar users/items to show similar scores inside the provided communities, this is not exactly true for the MovieLens dataset. We followed [Monti et al. 2017 and Rao et al. 2015] constructing the users/items graphs as 10-NN graphs in the space of user/items features (e.g. age, gender, occupation of users; and gender, year, etc. of the movies). The macro-communities in the users/items graphs built in this way do not necessarily coincides with clusters of similar values in the score matrix.
> >
> > A better alternative, which we did not explore in this paper, would be to construct the graphs from the data. Even better (a future direction mentioned in the response to Reviewer1) would be to learn the graph (or more specifically, the metric defined in this case on the feature space of the users, which determines the edge weights) together with the filters. We believe that this will allow to construct graphs where community structures are consistent with the data and thus result in a better performance.

---

### Official Review · AnonReviewer1 · 2017-11-28
**We find this work is interesting, timely, and of good quality to be presented in ICLR**

**Rating:** 8
**Confidence:** 3

**Review:**

This paper is on construction graph CNN using spectral techniques. The originality of this work is the use of Cayley polynomials to compute spectral filters on graphs, related to the work of Defferrard et al. (2016) and Monto et al. (2017) where Chebyshev filters were used. Theoretical and experimental results show the relevance of the Cayley polynomials as filters for graph CNN.

The paper is well written, and connections to related works are highlighted. We recommend the authors to talk about some future work.

---

> ### Author Response · Authors · 2017-12-20
> **Future work**
>
> We thank the Reviewer for a positive evaluation of our work.
>
> Future work: One of the key issues in our method (and deep learning on graphs in general) is the assumption of a given graph. In many settings, such as recommender systems, the graph has to be estimated from the data/side information. Learning the graph together with the filters on the graph is the next logical step which we will address in future works. In particular, for graphs constructed in some feature space (e.g. demographic information of users in the recommender system examples), “learning the graph” boils down to learning a metric on the feature space, which in turn determines the graph edge weights.
>
> Second, as we note in the response to Reviewer3, the behavior of our approximate matrix inversion is akin to “model compression”. In future work, we will analyze this phenomenon in light of previous  results on learnable iterative algorithms.

---

### Public Comment · (anonymous) · 2017-12-31
**unconvincing experiment results and marginally improvement**

It is well known that for the problem of graph vertex classification problem, standard data splitting for PubMed, Cora, and Citesser data sets are used as in [1-4] listed below. However, the authors only chose Cora data set, which is the smallest one of the three. Also, the authors use a data splitting method with much more labeled data. It makes the experiment unconvincing and difficult to compare performance with other papers.

Also, as the authors show in the paper, their improvement is marginal even using data splitting method defined by themselves compared with that in [2]. Please be noticed that [2] only uses a filter of adjacency matrix with order 1, however, this paper needs Laplacian matrix polynomials with order 12. The computational complexity is much larger than that of [2], and in this case, the marginal performance improvement could be omitted.

1. Revisiting semi-supervised learning with graph embeddings. ICLR 2016.

2. Semi-supervised classification with graph convolutional networks.
ICLR 2017.

3. Convolutional neural networks ¨on graphs with fast localized spectral filtering, NIPS2016

4 Geometric deep learning on graphs and manifolds using mixture model CNNs. CVPR 2017.

---

> ### Author Response · Authors · 2017-12-31
> **Standard data splitting and overfitting**
>
> We thank the reader for the provided comment.
>
> We are well aware of the data splitting outlined in [1-4]. However, we stress how our work is not aimed at outperforming the mentioned works in extreme semi-supervised learning problems. In particular, the solution outlined in [2] by Kipf & Welling is nothing more than a pure simplification of the architecture presented by Defferrard et al. at NIPS 2016, which dramatically reduces the amount of required parameters in order to cope with the small amount of available training samples* (140 in the standard splitting outlined in [1-4] for the CORA dataset). Since however the main term of comparison is represented in our case by ChebNet, we decided to extend the amount of available data in order to avoid overfitting and thus exploit more complicated but powerful models. This shows in particular how if a sufficient amount of training samples are provided, the approach outlined in [2] appears as just a suboptimal solution able to achieve non-optimal filters because of the simplicity of the defined operator (in the end convolution in [2] is nothing more than a weighted sum of the features available in the one-hop neighborhood that by no means really exploits the local topology of the provided domain**). Please, see in this sense Figure 5 left of our paper where we show how both ChebNet and CayleyNet are able to outpeform GCN by respectively 0.6% and 1.4% (i.e. not a marginal improvement).
>
> We will provide our data splitting whether the paper will be accepted in order to provide a valuable term of comparison to the community.
>
> * this is in general the main reason why the authors in [2] don't compare ChebNet with the proposed architecture.
> ** To see this let's consider the MNIST classification problem outlined in ChebNet and our paper. If instead of ChebNet/MoNet/CayleyNet, GCN would be used, all the features produced at the various convolutional layers would be nothing more than just simple averages of the original grey levels available in the provided pixels i.e. not meaningful representations of the local image behavior. GCN appears in this sense as a good solution whenever a small amount of training data is available (and multiple input features), however richer solutions can be exploited to better discriminate interesting local patterns if a sufficient amount of labeled samples can be provided  (i.e. as in the proposed CORA experiment).

---

> > ### Public Comment · (anonymous) · 2018-01-03
> > **cannot work for semi-supervised learning problem**
> >
> > According to your reply, can I conclude that CAYLEYNETS cannot work for semi-supervised learning studied in [2]?
> > Also, with more labeled data in CORA, the improvement is still marginal compared with GCN in [2] as shown in your experiment.

---

> > > ### Author Response · Authors · 2018-01-04
> > > **cayleynet and extreme semi-supervised learning problem**
> > >
> > > As we already stated, CayleyNet (as ChebNet, which is our main term of comparison) is not meant to work with the extreme conditions depicted in [2].
> > >
> > > Also, stating that CayleyNet "cannot work for semi-supervised learning problem" is misleading. CayleyNet outperforms all the competitors on the semi-supervised learning problem we considered, achieving an 87.9% of accuracy with order 1 and symmetric normalized laplacian compared with the 86.50% of GCN.

---

### Public Comment · (anonymous) · 2018-01-01
**High computational complexity for large-scale graph**

This paper needs Laplacian matrix polynomials with order 12, which means the computational complexity is O(12N^3), with N being the number of graph nodes. The computational complexity is overwhelming even consider a small graph with N=10,000 nodes. With such high computational complexity, the performance improvement compared with other method is marginal.

---

> ### Author Response · Authors · 2018-01-02
> **Computational Complexity**
>
> As already presented in Section 3 and Fig. 4 center-right, the computational complexity of the proposed method scales linearly wrt number of vertices available in the given domain for sparse graphs (and thus complexity is a O(n) and not a O(n^3)). Furthermore, stating that the method “needs Laplacian matrix polynomials with order 12” is erroneous and misleading. The number of Jacobi iterations required by CayleyNet is problem dependent and as we shown in our community detection experiment (Fig.4 left) even a small amount of iterations (e.g. 1-5) may be sufficient for significantly outperforming the performance achieved by ChebNet.

---

> > ### Public Comment · (anonymous) · 2018-01-02
> > **High computational complexity as need to compute Eigenvalues of Graph Laplacian**
> >
> > As in Section 3, eigenvalues of Graph Laplacian is needed. However, computing the eigenvalues leads to additional operation with complexity O(N^3).

---

> > > ### Author Response · Authors · 2018-01-02
> > > **Unnecessary Eigendecomposition**
> > >
> > > This statement is (again) wrong. We highly recommend the anonymous commenter to familiarize him/herself with previous works on spectral graph CNNs and also carefully read our paper.
> > >
> > > Unlike the first paper on spectral CNNs by Bruna et al where explicit eigendecomposition of the Laplacian is performed, the main point of the follow up works (Defferrard et al, Kipf&Welling and our present paper, which is based on the former) is to avoid  this expensive operation altogether.
> > >
> > > The way to do it is to implement a filter as a function of the Laplacian f(Delta)*x applied to the graph signal x. This is equivalent to applying f to the Laplacian eigenvalues, but does not require their explicit computation if f can be expressed in terms of simple matrix operations (addition, multiplication, and scaling).
> > >
> > > Defferrard et al used polynomial functions as f. In this case, the resulting filter is an FIR in signal processing terms and it’s computation amounts to multiplying the signal by the Laplacian matrix r times, where r is the polynomial degree. If the graph is sparsely connected with O(n) edges, this costs O(n) operations.
> > >
> > > In our paper, we use rational functions as f, which are IIR filters. The whole point of our paper is how to compute such functions efficiently with linear complexity. The use of Jacobi iterations again brings the computation to a series of Laplacian multiplications which has O(n) complexity.

---

> > > > ### Public Comment · (anonymous) · 2018-01-03
> > > > **Complexity differences between different orders of matrix polynomials**
> > > >
> > > > As in your reply, I quote " it’s computation amounts to multiplying the signal by the Laplacian matrix r times...If the graph is sparsely connected with O(n) edges, this costs O(n) operations. "
> > > >
> > > > I am confused now as you said the computational complexity is O(n), which has no relationship with the order r you choose? How can that be? How do you define the sparsity of a graph? If a graph is not that "sparse", what is the computational complexity?

---

> > > > > ### Author Response · Authors · 2018-01-03
> > > > > **Computational complexity and order r**
> > > > >
> > > > > The order r is an architecture parameter independent of the input size, hence r=O(1). Typically, it's a small number 1-10. Therefore, it is just a constant in the complexity.

---

### Author Response · Authors · 2018-01-04
**Revision**

Changes in the new version:
On page 5, the complexity section was updated, and gives more emphasis on the trade-off in the choice of parameters.
On page 6, the section "Chebyshev as a special case of Cayley" was added.
On page 7, in the MNIST experiment, the number of Jacobi iterations and the run-times were made explicit.
On page 8, Table 2 was updated to compare ChebNets and CayleyNets based on the same number of real coefficients.
On page 8, in "Citation network", the nonormalized Laplacian was replaced by the scaled nonormalized Laplacian, and additional experiments were added to compare ChebNets and CayleyNets with the same number of parameters. Figure 5 was thus updated.
On page 9, in "Recommender system", the number of parameters in the ChebNet was updated to match the number of parameters in the CayleyNet. A comparison of the run-times was also added.
On page 14, a computational complexity appendix was added, that better compares run-times of ChebNets and CayleyNets.

---

### Decision · Program_Chairs · 2018-01-29
**ICLR 2018 Conference Acceptance Decision**

**Decision:**

Reject

**Comment:**

This paper considers graph neural representations that use Cayley polynomials of the graph Laplacian as generators. These polynomials offer better frequency localization than Chebyshev polynomials. The authors illustrate the advantages of Cayleynets on several benchmarks, producing modest improvements.

Reviewers were mixed in the assessment of this work, highlighting on the one hand the good quality of the presentation and the theoretical background, but on the other hand skeptical about the experimental section significance. In particular, some concerns were centered about the analysis of complexity of Cayley versus the existing alternatives.

Overall, the AC believes this paper is perhaps more suited to an audience more savvy in signal processing than ICLR, which may fail to appreciate the contributions.